# Do socioeconomic inequalities contribute to the high prevalence of child developmental risk in an ethnically diverse, socioeconomically disadvantaged population? A Born in Bradford's Better Start (BiBBS) study

Kate E Mooney [1,2] Josie Dickerson,[2] Sarah Louise Blower,[1] Matthew Walker,[2] Jennie Lister,[1] Kate E Pickett [1]

¹Health Sciences, University of York, York, UK
²Bradford Institute for Health Research, Bradford, UK

**Correspondence to**
Dr Kate E Mooney; kate.mooney@york.ac.uk

## ABSTRACT

**Background** Socioeconomic inequalities in child development are pervasive; however, less is known regarding the impacts of socioeconomic factors within and across ethnically diverse and socioeconomically disadvantaged populations. This study (1) describes the prevalence of children at risk of poor overall early child development; (2) investigates the relationship between individual indicators of socioeconomic position and early child development; and (3) investigates if the relationship between indicators of socioeconomic position and early child development varies by ethnic group.

**Methods** This study uses data from a prospective birth cohort study, Born in Bradford's Better Start (BiBBS). Child development was measured with the Ages and Stages Questionnaire (ASQ) during routine health visiting appointments at age 2-years-old. Binary logistic regression investigated child development by key maternal socioeconomic indicators: maternal education, financial security, social status (measured via the MacArthur Scale of Subjective Social Status), and social support (measured via number of people to count on).

**Results** 22% of the 2003 children with a valid developmental assessment were at risk of poor child development. Mothers who had a degree (OR=1.95, 95% CI 1.28 to 2.99), reported 'living comfortably' in financial security (OR=1.78, 95% CI 1.03 to 3.07) and had higher social status (OR=1.11, 1.02 to 1.22); all had higher odds of their child having a good development. Though socioeconomic gradients in maternal education and financial security were consistent across White British, South Asian and Other ethnic groups, both social support and social status had weaker relationships with child development for South Asian parents.

**Conclusion** A high proportion of children are at risk of poor development in this diverse, socioeconomically disadvantaged population. Higher socioeconomic position may protect against poor early development, and the mechanisms underlying this may differ by ethnicity. The findings underline the need for proportionate universal strategies to improve child development in such communities.

## WHAT IS ALREADY KNOWN ON THIS TOPIC

⇒ Socioeconomic inequalities in early child development are pervasive.
⇒ While objective socioeconomic indicators such as higher parental education are established predictors of child development, less is known regarding subjective indicators such as social support and subjective social status, particularly within ethnically diverse populations.

## WHAT THIS STUDY ADDS

⇒ A high proportion (22%) of children in the socioeconomically disadvantaged and ethnically diverse Born in Bradford's Better Start (BiBBS) study population are at risk of poor child development.
⇒ Higher maternal education, financial security and subjective social status are associated with better child development, with the steepest socioeconomic gradient by subjective social status.
⇒ The relationships between social support and subjective social status with child development varied by ethnic group, as these did not relate to child development for South Asian parents.

## INTRODUCTION

The first 3 years of life are a critical period for childhood development, which encompasses the development of physical, mental, social and emotional capabilities.[1 2] Early childhood development promotes future educational achievement and health,[3 4] and persistent difficulties with early cognitive and socio-emotional development can lead to worse physical and mental health in late adolescence.[5] Prevention and early intervention to

**HOW THIS STUDY MIGHT AFFECT RESEARCH, PRACTICE OR POLICY**

⇒ The BiBBS cohort can be used to further understand patterns of childhood inequalities in socioeconomically deprived, ethnically diverse populations.

⇒ Future research should focus on how and why various socioeconomic factors impact child development within socioeconomically deprived, ethnically diverse populations.

⇒ The psychological impacts associated with low subjective social status could be addressed through participation in group-based parenting programmes.

⇒ A proportionate universal approach, with both universal support available to all parents and targeted intervention programmes for those in need of additional support, is recommended to improve children's outcomes.

promote strong early development for all children may reduce socioeconomic disparities across the lifespan.[6]

## Socioeconomic position and child development

Health inequalities are unfair and avoidable differences related to personal, demographic and socioeconomic characteristics. The majority of the relevant literature demonstrating pervasive socioeconomic inequalities in early child development in the UK is based on cohort studies, across various socioeconomic exposures and multiple domains of child development.[7] The Millennium Cohort Study (MCS) (n=19 517) is a nationally representative sample of children born between 2000 and 2002. In MCS, outcomes examined at age 3 include socio-emotional difficulties measured via the Strengths and Difficulties Questionnaire (SDQ) and cognitive development measured via the Bracken Basic Concept Scale. At age 5, socio-emotional difficulties were measured again via the SDQ and verbal ability via the British Ability Scales. In MCS, higher parental education predicted higher verbal ability at age 5,[8] and higher family socioeconomic position predicted lower socioemotional difficulties at ages 3 and 5.[9] Families with the lowest incomes had children with a higher prevalence of both socio-emotional and cognitive difficulties at both 3 years old and 5 years old.[10] A study using the Avon Longitudinal Study of Parents and Children (ALSPAC, n=13 855) similarly reported that lower parental occupation, education and financial security were associated with lower expressive and receptive language at 2 years old measured as a modification of the MacArthur Communicative Development Inventory (CDI).[11]

Socioeconomic position may impact children's development through multiple mechanisms, for instance, socioeconomic advantage may afford parents better resources and education for enhancing the home environment of children, including via educational inputs such as language, play and stimulation, and emotional inputs such as parent-child relationships and interaction.[12] On the other hand, socioeconomic disadvantage may have a negative impact on children's development through poor parental mental health and stress, impacting a parent's

engagement and relationship with their child.[12–14] Subjective social status captures a person's perception of their social class relative to other people and may have an impact on children's outcomes through a heightened experience of chronic stress responses.[15] It is therefore of interest to compare the magnitude of associations for multiple measures, including not only objective measures such as income and education, but also subjective aspects of societal position.[16] This can give insights into the mechanisms by which family socioeconomic position influences children's development.[17] Despite this, much of the previously summarised literature applies objective measures of socioeconomic position such as education, occupation or income.[8–11]

### Subjective measures

For socioeconomically deprived families, subjective measures may be more sensitive predictors of children's outcomes than measures of income, as they can more accurately assess whether lower-income families feel they can provide for basic needs.[17] For example, in the Born in Bradford cohort, parents' self-reported financial security more strongly predicted birth outcomes, in comparison to parental education, receipt of means-tested benefits and paternal occupation.[18] Further, subjective social status indicators, such as the Macarthur Scale of Subjective Social Status, may encourage consideration of perceptions of relative inequality, and this has been associated with cognitive function and self-rated health in adulthood.[19] Within the sample used in this study, most pregnant women rate themselves as having higher subjective social status than others in their neighbourhood, despite living in socioeconomically deprived areas.[15] While self-reported social status appears to be related to adult well-being,[20] it is yet to be investigated with regards to children's outcomes.

Parental social support refers to the existence of social networks.[21] Although social support is not directly a measure of a family's socioeconomic position, it can be considered a specific manifestation of social capital, which refers to the resources that are obtained via a person's social networks.[22 23] Parental social support could alleviate challenges during the transition to parenthood, hence supporting childhood outcomes,[21] and could potentially mitigate against the negative effects of socioeconomic disadvantage for socioeconomically disadvantaged families. Emerging evidence suggests that higher levels of social support during pregnancy are associated with improved cognitive development at age 2[24] and lower prevalence of developmental delay at age 3.[25]

### Ethnicity and child development

As ethnic minority groups tend to experience higher levels of socioeconomic disadvantage in England,[26] a consideration of differences in child development by ethnicity might reveal differential impacts of socioeconomic position. Indeed, an MCS study revealed that most ethnic minority groups have lower scores in cognitive and

socio-emotional outcomes compared with White British children.[12] A study of routinely collected health visiting data between 2018 and 2021, with a sample of 226 505 children at age 2–2 ½ years, found that 86% of children met expected levels of development overall, but that children from the most deprived neighbourhoods (82.6%) and children recorded as Black ethnicity (78.9%) were less likely to meet expected level of development.[27]

### Socioeconomic gradients within ethnic groups

Previous research has demonstrated that socioeconomic differences appear to be less pronounced in ethnic minority groups than ethnic majority groups for maternal and child health outcomes when comparing Pakistani-born to UK-born mothers,[18 28] child socioemotional difficulties when comparing White to ethnic minority groups[29] and working memory when comparing Pakistani to White British children.[30] This lack of socioeconomic gradient may be due to difficulties with measurement, for instance, inaccurate measurements of socioeconomic position in ethnic minority groups (eg, educational attainment if received in a different country) or smaller samples in minority groups being underpowered to detect associations. Or, these findings may be due to a true lack of relationship between socioeconomic position and children's outcomes for ethnic minority groups. Subjective measures of socioeconomic position may aid our understanding of relationships between socioeconomic position and child outcomes for ethnic minority groups, as they may be more relevant than other measures (e.g. education), and may also provide an insight into the mechanisms by which minority families mitigate against the negative effects of disadvantage. Though the relationship between ethnicity and children's outcomes is likely to be influenced by a variety of other factors beyond socioeconomic position, such as differences in culture and experiences of prejudice, discrimination and racism.[31] Relevant to the sample included in this study, South Asian families' parenting styles may differ from other ethnic groups, and South Asian families are more likely to involve intergenerational or extended support in parenting itself.[32–34] The relationship between socioeconomic experiences and children's outcomes may therefore differ between ethnic groups, and this relationship may be mitigated or exacerbated by different cultural practices, such as parenting. Hence, larger samples of ethnic minorities and a variety of socioeconomic measures are needed to investigate the true impacts of socioeconomic position across diverse ethnic groups.

### Present study and rationale

The present study uses data from the Born in Bradford's Better Start (BiBBS) interventional birth cohort, which has recruited a representative sample of 5758 pregnant mothers and their children living in three inner-city areas of Bradford. BiBBS recruited families throughout the time period that 'Better Start' interventions were being delivered (1st January 2016 to 31st July 2024).

While BiBBS was specifically designed to undertake efficient and pragmatic evaluations of these multiple early life interventions, the cohort data can also be used for understanding how inequalities develop in disadvantaged populations. The mothers in the BiBBS cohort are predominantly from ethnic minority backgrounds (88%) and live in the most deprived decile of the Index of Multiple Deprivation (84%).[35]

This under-researched population provides the opportunity to understand potential differences in children's development by both subjective and objective socioeconomic measures, enabling an investigation of whether socioeconomic differences exist within disadvantaged contexts and among different ethnic groups. This may also provide insights into which factors may protect against the development of poor childhood development in a predominantly socioeconomically disadvantaged population, and which groups of children may need more support. Building on previous research which has tended to focus on one specific aspect of child development at a time, we use the Ages and Stages Questionnaire, which is a broad measure of child development that encompasses communication, fine and gross motor skills, problem solving and personal-social development. In this study, we:

1. Describe the prevalence of children at risk of poor child development overall.
2. Investigate which individual indicators of socioeconomic position predict early child development.
3. Investigate if the association between individual indicators of socioeconomic position and early child development varies by ethnic group.

## METHODS

### Study design

This is a secondary analysis of observational longitudinal cohort study data. We have used the Strengthening the Reporting of Observational Studies in Epidemiology (STROBE) (2021) guidelines for reporting methods (see online supplemental file A).[36]

### Study setting

This study is set in Bradford, a city in Northern England with high levels of socioeconomic deprivation and a large ethnic minority population.[37] Part of the Born in Bradford (BiB) research programme, Born in Bradford's Better Start (BiBBS) captures in-depth baseline data during pregnancy, and participants provide written consent for routine linkage to their own and their children's health, education and intervention participation records. Pregnant women were invited to join BiBBS by trained researchers in person. Recruitment took place predominantly at Bradford Royal Infirmary's Glucose Tolerance Test (GTT) clinic at 24–28 weeks gestation, which is offered to all women booked for delivery at BRI, and secondary recruitment sources in the community.[35 38] An interim BiBBS cohort profile for mothers

recruited between 2016 and 2020 was shown to be broadly representative of the pregnant population in terms of ethnicity, parent age, parity, language ability and area deprivation.[35]

## Data

All predictive measures were measured during the BiBBS baseline questionnaire, administered when mothers were approximately 26 weeks pregnant with their child. We linked Health Visiting records to provide the Ages and Stages Questionnaire (ASQ) outcome for this study which was collected approximately 2–2.5 years post birth. Inclusion for this study was all recruited BiBBS pregnancies between January 2016 and December 2022, who had an ASQ recorded for that pregnancy at the 2–2 ½ year visit by October 2024 (the time of the data drop from the data source).

Inclusion of covariates was decided using a Directed Acyclic Graph (DAG) (see below), and the analytical form of variables is described below.

## Measures

We used four indicators of socioeconomic position. First, maternal education was categorised into 'No qualifications', 'five or less General Certificate of Secondary Education (GCSEs)', 'five or more GCSEs', 'A levels' and 'Degree'. Second, financial security was ascertained by mother's response to "how well are you managing financially?" Responses were coded as: 'Just about getting by', 'Finding it quite/very difficult', 'Doing alright', 'Living comfortably'. Third, subjective social status was measured via the MacArthur Scale of Subjective Social Status, an image of a ladder with 10 rungs, numbered from one at the bottom to 10 at the top. We used mother's responses to "Where would you place yourself on the ladder in relation to other people in your neighbourhood?", with answers analysed as continuous, ranging between 0 and 10. Fourth, social support was measured via responses to "how many people in your life can you count on in times of need?", again analysed as a continuous predictor, and answers ranged from 0 to 10. To test the sensitivity of the results to the linearity assumption, we estimated additional models with subjective social status and social support estimated as categorical variables, categorised into low (0–2 people, 1–2 ladder), medium (3–5 people, 3–8 ladder) and high (6 to 10 people, 9–10 ladder). Any 'do not wish to answer' and 'don't know' responses were recoded as missing. The lowest socioeconomic category was used as the reference category.

For objective 2, maternal ethnicity was grouped into the largest ethnic groups within the cohort: White British, Pakistani heritage, White Other (containing White Irish, Polish, Slovakian, Romanian, Czech, Other White and Gypsy/Roma/Irish Traveller), Other South Asian (Indian and Bangladeshi) and Other (Chinese, African, Caribbean, Mixed or Other). For objective 3, ethnic categories were further collapsed. Hence, the groups were White British, South Asian (Pakistani, Indian and Bangladeshi)

and Other. South Asian ethnicities were grouped as one due to potential cultural similarities in parenting style among these groups,[32–34] and similarities in historic migration patterns.[39] We grouped the 'Other' ethnicities and retained them as (1) the sample size of the ethnic minority groups was too small to estimate separately, and (2) such groups have historically been excluded from research and analysis.[40] Hence, although the 'Other' group contains multiple heterogeneous ethnic groups with a variety of experiences, including their data may allow for an insight into patterns in their outcomes and generate further hypothesis testing with larger samples. White British were the reference group.

There were three binary covariates: (1) Maternal country of birth was analysed as Not born in the UK/Born in the UK, (2) Maternal first spoken language was analysed as Not English/English and (3) Child sex was analysed as Female/Male. There was one continuous covariate, (4) child age in months at the date of the most recent ASQ.

## Outcome measure (Child development via Ages and Stages Questionnaire (ASQ))

All parents living in England receive a visit from their health visitor when their child turns 2 years old, during which children's development is assessed using the 24, 27 or 30 month Ages and Stages Questionnaire (ASQ). ASQ-3 was developed to screen for developmental delay and comprises 21 age-specific questionnaires for children aged between 1 month and 66 months.[41] ASQ-3 covers five key domains of developmental status of a child's following areas:[42] Communication: babbling, vocalising, listening and understanding; gross motor: arm, body and leg movements; fine motor: hand and finger movements; problem solving: learning and playing with toys and personal-social: solitary social play and play with toys and other children. Several studies have shown the ASQ-3 to be a valid screening tool to identify developmental concerns for individual children, though there is a lack of its validation in England specifically, and for ethnic minority groups.[43 44]

There are three response options to each question (yes/sometimes/not yet, with scores being yes=10, sometimes=5 and not yet=0). The score of each domain ranges between 0 and 60; however, each age version and domain of the ASQ has different cut-offs to categorise children into (1) 'below cut-off', where a child requires further assessment and intervention with a professional, (2) 'monitor', where a child's development should be monitored, and (3) 'above cut-off', where a child's development is on schedule. For objective 2, in line with the Public Health England method for calculating whether children have a Good Level of Development (GLD), an overall ASQ score was created by categorising as 'Not at risk'=above cut-off and/or monitor on all five domains, and 'At risk'='below cut-off' on one or more domains (see https://fingertips.phe.org.uk/static-reports/health-trends-in-england/England/best_start_in_life.html[27]).

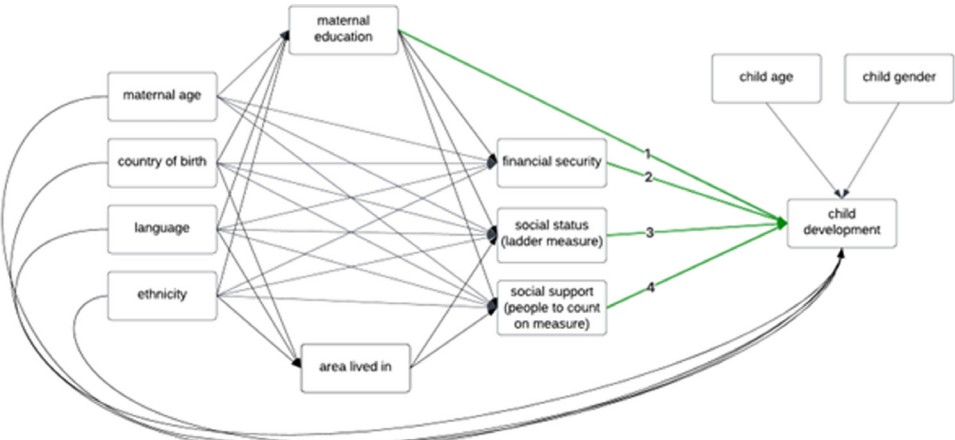

**Figure 1** Directed Acyclic Graph with estimates of interest labelled 1–4.

For objective 3, to create an outcome that was more sensitive to identifying children *potentially* at risk, an additional ASQ 'Full' GLD (FGLD) score was created by categorising as 'Not at risk'='above cut-off' on all five domains and 'At risk'='below cut-off' AND/OR 'monitor' on one or more domains. Scores were only calculated if all five domains were available. The FGLD method provides more variation in the outcome than the GLD method, which means it is more sensitive to identifying any existing socioeconomic gradients in outcomes.

## Analysis

For objective 1, we described the number of children at risk of poor development overall using both the GLD and FGLD method. For reference, we compared the proportions of children at risk using the GLD method to Jung et al.[27], who described child developmental outcomes in the ASQ-3 across England.

For objectives 2 and 3, the Directed Acyclic Graph (DAG) presented in figure 1 ascertained the adjustment set for the effect of each socioeconomic indicator (maternal education, financial management, social ladder and number of people to count on). Socioeconomic variables could be either mediators or confounders of specific associations depending on the relationship of interest, hence each relationship of interest required a separate adjustment set and analysis.[45] Justification for inclusion of each confounder/covariate was based on prior literature.[10 12] We did not

include area deprivation and/or area as a covariate, since the areas under study are relatively homogeneous in terms of deprivation.[35]

For objective 1, the prevalence of children at risk of poor development was described using the GLD method. For objectives 2–3, binary logistic regression models were conducted to produce ORs and 95% CIs using the FGLD method, and model-based estimates (ie, adjusted predictions) for the outcomes were presented using plots.

For objective 3, to investigate if socioeconomic differences in child development varied by ethnic group, we estimated four additional binary logistic models with ethnicity (categorised as White British, South Asian and Other) and each socioeconomic predictor (maternal education, financial management, social ladder and people to count on), with an interaction fitted between these variables. We applied the specified adjustment sets for each predictor using table 1. A likelihood ratio test was performed to compare the fit of the models with and without the interaction by ethnic group, where the fit of the two competing models was compared and the statistic represents a direct comparison of the relative likelihood of the data, given the best fit of the two models.[46]

Results were considered to be statistically significant if p<0.05. R was used for all analysis, key packages included *tidyverse, ggplot2, ggdist* and *ggeffects*. Full analysis code and results are available at https://osf.io/c9rup.

**Table 1** Model number with key predictor, and covariates included in each model

| Model (#) and key predictor | Confounders included in model | Covariates included in model |
|---|---|---|
| Maternal education | Migrant to UK, English is the first language, Ethnic group | Child age, child sex |
| Financial management | Maternal education, Migrant to UK, English is the first language, Ethnic group | Child age, child sex |
| Subjective social status (social ladder) | Maternal education, Migrant to UK, English is the first language, Ethnic group | Child age, child sex |
| Social support (people to count on) | Maternal education, Migrant to UK, English is the first language, Ethnic group | Child age, child sex |

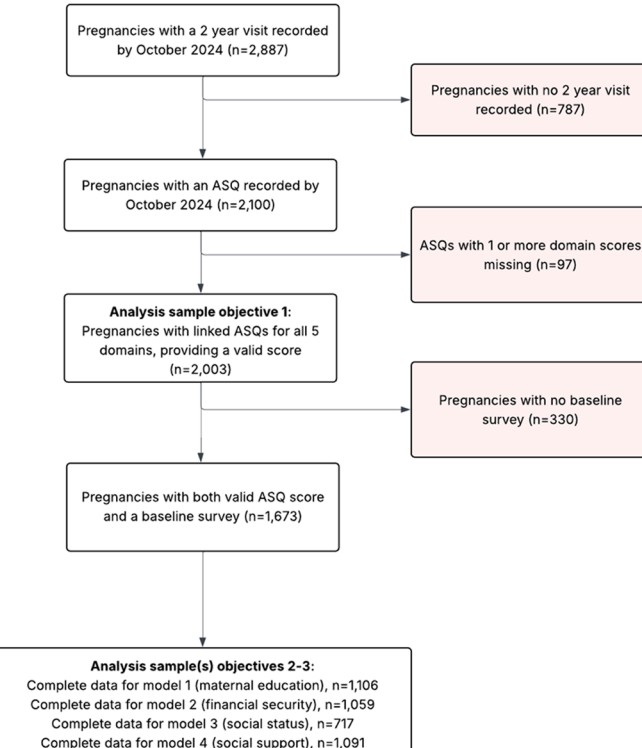

**Figure 2** Eligible participants and numbers included in models for each objective. ASQ, Ages and Stages Questionnaire.

## Missing data

We applied complete case analysis, as this gives unbiased results when the chance of being a complete case does not depend on the outcome after taking the covariates into consideration.[47] The analysis therefore assumes that having complete data does not depend on a child's development, after taking into account the included predictor and control variables.

## Patient and public involvement

Members of the community are involved in the design and conduct of BiBBS cohort research, including recruitment, questionnaires, measures and interpretation and dissemination of findings, through regular Community Research Advisory Group meetings, composed of Better Start Bradford residents.

## RESULTS

Full analysis code and results are available at https://osf.io/c9rup.

## Sample

Figure 2 shows the number of pregnancies with ASQ records and the sample sizes for each objective and model. The sample contained 2887 pregnancies with a 2-year visit recorded, and 2100 (72%) of them had an ASQ recorded. It is beyond the scope of this current study to explore reasons behind non-completion of the ASQ; however, a mixed methods study by our team will

explore this further.[48] The analysis sample for objective 1 contains all pregnancies with valid ASQs (n=2003), and the analysis sample for objectives 2–3 depends on the availability of the data within the sample (social status n=791, financial security n=1159, social support n=1191 and maternal education n=1214).

## Objective 1 (prevalence of children at risk)

Objective 1 was to estimate the number of children at risk of poor child development overall, and the number of children at risk of poor development in each domain. Within the 2003 individual children with linked ASQs, 445 (22%) of all children were at risk of overall poor development using the GLD method, and 884 (44%) were at risk using the FGLD method.

## Objective 2 (which socioeconomic indicators predict child development)

Objective two was to investigate the association between individual indicators of socioeconomic position and early child development. The FGLD method was used as the outcome, which is more sensitive to identifying children potentially at risk of poor development (44%). Descriptives for all participants are presented in online supplemental file B, and logistic regression models are presented in online supplemental file C.

ORs from adjusted logistic regressions between the predictor variables and achieving the expected level of development show that having a degree (OR=1.95, 95% CI 1.28 to 2.99), living comfortably (OR=1.78, 95% CI 1.03 to 3.07) and subjective social status (OR=1.11, 95% CI 1.02 to 1.22), all resulted in increased odds of having a FGLD. Social support was not associated with having a FGLD (OR=0.99, 95% CI 0.94 to 1.04).

All socioeconomic predictors, except for social support, were significantly associated with the probability of expected development, and the figures visualise the socioeconomic gradients. Figure 3a–d presents the adjusted predictions, showing that the socioeconomic gradient is steepest by subjective social status, with the largest difference between the highest and lowest socioeconomic category for this predictor.

## Objective 3 (variation between socioeconomic position and child development by ethnic group)

Objective 3 was to investigate if the association between individual indicators of socioeconomic position and early child development varies by ethnic group. Results from the likelihood ratio tests indicated that the model with the interaction fitted the data equally well for maternal education ($X^2$=8.22, p=0.222) and financial security ($X^2$=4.34, p=0.461), whereas the model with the interaction was superior for subjective social status ($X^2$=2.90, p<0.001) and social support ($X^2$=6.26, p<0.001). Figure 4a–d presents the adjusted predictions, indicating that subjective social status and support were less strongly related to child development for South Asian groups, in comparison to White British and Other ethnic groups.

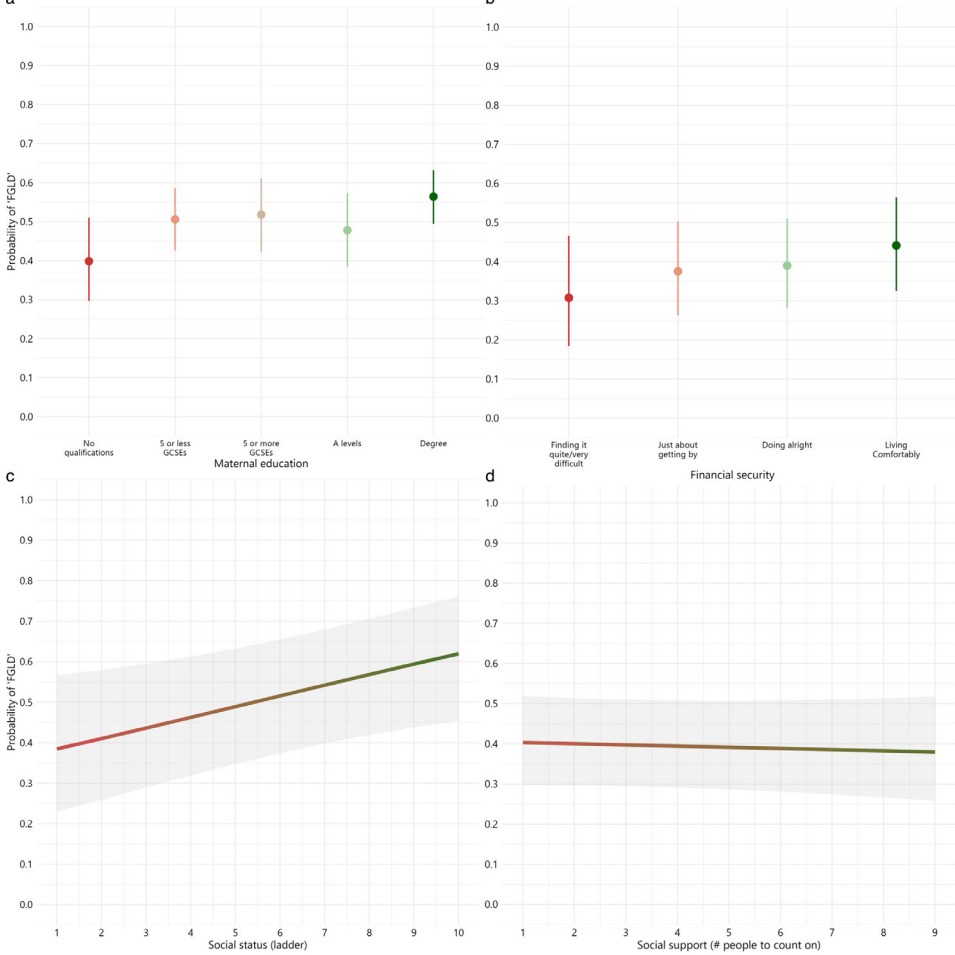

**Figure 3** (a–d) Adjusted predictions for models 1–4, with no interaction for ethnicity included. Note: the overlap of 95% CIs in a and b does not mean that the difference between any given pair of estimates is not significant at p<0.05. FGLD, full good level of development.

Full model results are in online supplemental file D, and adjusted predictions from each model are presented below. The sensitivity analysis, including subjective social status and support as categorical predictors, showed a similar pattern of results.

## DISCUSSION

Our analysis of longitudinal, routinely collected data from this place-based birth cohort study reveals a concerningly high proportion of children in inner-city Bradford identified as being at risk of poor development by age two (22%). The prevalence of children at risk is considerably higher than a previously estimated average for England (14%),[27] underscoring the significant vulnerability of this specific population. This high rate of developmental risk will likely have long-lasting implications for these children's future educational attainment, health and overall well-being.[6]

While maternal education and financial security had previously been established as important predictors of children's outcomes,[7 18] our study shows the additional relevance of parent-reported social status. Subjective

social status presented the steepest social gradient in children's outcomes in comparison to maternal education (18 percentage point gap), financial security (15 percentage point gap) and social support (2 percentage point gap), with a 26% point gap between the highest and lowest points on the subjective social status scale. While research linking subjective social status to early developmental outcomes is limited, this aligns with broader research indicating that subjective socioeconomic measures may more strongly predict birth outcomes in deprived populations, when compared with parental education.[18] This suggests that a self-reported, subjective assessment of social standing offers a more nuanced understanding of socioeconomic position within disadvantaged populations, potentially revealing how parents in deprived areas might mitigate the challenges of socioeconomic deprivation through their own perceived subjective social status.

Since this measure of subjective social status purportedly captures a person's perception of their social class relative to other people, this may indicate that a parent's perceptions of inequality have an impact on their

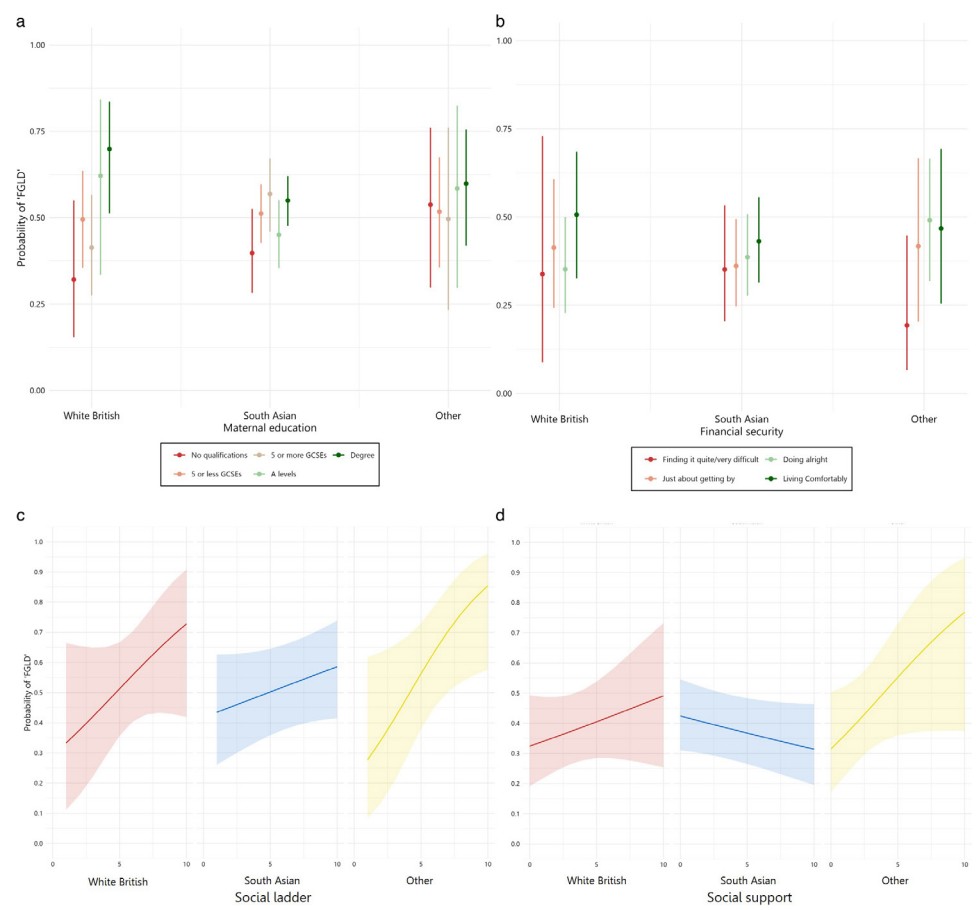

**Figure 4** (a–d) Adjusted predictions for models 1–4, with interaction by ethnicity included. FGLD, full good level of development.

children's outcomes through a heightened experience of chronic stress.[15 49] Since few epidemiological studies include both subjective and objective measures of socioeconomic position, further research is necessary. We propose that subjective socioeconomic measures may be particularly useful in ethnically diverse populations through potentially obtaining lower non-response rates compared with objective metrics (like exact income or foreign qualifications) that can be challenging to reliably report. Further research is needed regarding the validity and generalisability of socioeconomic measures across different ethnic groups, and their relationships with various outcomes. The lack of a significant association between social support and child outcomes contrasted with previous research that has indicated a positive relationship;[24 25] however, further analysis revealed that this may be due to differences by ethnic group.

While the socioeconomic gradients in maternal education and financial management appeared broadly consistent across White British, South Asian and Other ethnic groups, a more nuanced picture emerged regarding the role of subjective social status and support. The adjusted predictions (figure 4a–d) revealed a relatively weaker relationship between subjective social status and support and child outcomes specifically within the South Asian group, (in comparison to the White British and Other ethnic

groups). This suggests that the null result in the original social support model was perhaps driven by the weak association for South Asian groups, suggesting social support may be a protective factor for many families. Interestingly, a very strong association was apparent between social support and child development for the 'Other' ethnic group, which includes White Other (including White Irish, Polish, Slovakian, Romanian, Czech, Other White, Gypsy/Roma/Irish Traveller), Black African/Caribbean/Black British and Mixed ethnic groups. As these groups represent the least populous ethnic minorities in the BiBBS population, it may be that social support is more important for these communities. However, it is crucial to acknowledge that the 'Other' ethnic group in this study is heterogenous and hence there may be hidden variation across ethnicities in the association between social support and child outcomes. Hence, further research is needed with these smaller ethnic minorities to explore these associations in depth. The South Asian population is the most populous ethnic group within the BiBBS area, perhaps meaning that they benefit from the psychosocial resilience offered by same-ethnic density through shared culture, networks and social capital. This may mean that the volume of social support at a more individual level is not as crucial for improving child outcomes.[50 51] The lack of association for South Asian groups aligns with existing

research that has reported a lack of expected socioeconomic gradients for maternal and child outcomes among South Asian groups,[18 28 30] though these previous studies used different socioeconomic indicators. Our study extends this by demonstrating that a subjective measure of social support may have different relationships with children's outcomes depending on ethnic group, and may be particularly important for less populous ethnic minorities.

However, this finding could also reflect unmeasured confounding or inaccurate measurement. For instance, our measure of social support might not capture the *quality* or *type* of support received, which could be more influential than the quantity of social connections. Given that South Asian families in this area are often characterised by close proximity to family members living in the same households,[52] the nature and dynamics of this support might differ significantly from other groups. These strong familial networks could potentially buffer against the negative effects of socioeconomic disadvantage, though research in this area is lacking.

Altogether, the strong relationship between subjective social status and child outcomes may have implications for early interventions that support families facing socioeconomic disadvantage. This may suggest that early parenting interventions should not only provide educational material and financial aid to families, but also address the psychological impacts associated with low subjective social status.[49] This indicates the importance of the social environment for parents of young children, highlighting the role that group-based parenting programmes could play in building social networks and social cohesion for parents of young children.[53] However, as we found that the importance of subjective social status and support may vary by ethnic group, further research on this topic is needed with diverse ethnic groups.

Ultimately, the relationship between ethnicity and children's outcomes is of course influenced by a multitude of factors beyond socioeconomic position, such as cultural practices, prejudice and racism.[31] Early years practitioners working with families should be mindful of potential cultural nuances in how social factors influence child development within different communities. To gain a more comprehensive understanding of the true impact of socioeconomic position across diverse ethnic groups, future research should continue to employ larger samples of ethnic minorities and apply a wider array of both objective and subjective socioeconomic measures.

## Strengths and limitations

Findings from BiBBS may generalise to many similar urban populations, especially those with high ethnic diversity and socioeconomic deprivation. The large number of Pakistani families recruited (e.g. n=979, 49% of the sample) improves our statistical power to detect differences and builds on previous research describing outcomes for this ethnic minority group, which may have previously been limited by smaller numbers of ethnic minorities. While it is a strength that our sample on the whole is ethnically diverse, we were limited in our ability to describe associations between socioeconomic position and child outcomes within each of these groups due to small group sizes. Further, the disadvantaged population under study may be relatively homogeneous in terms of socioeconomic characteristics, reducing the potential for observing significant social gradients. It was not possible to include income as a predictor in this study, as this was not included in the baseline questionnaire due to low acceptability in this population. Some families were missing a baseline questionnaire (around 25%) due to changes to recruitment procedures during COVID, and some children were missing the ASQ outcome (28%, see (Figure 2). Hence, our complete case analysis is only valid under the assumption that having complete data does not depend on a child's development, after taking into account the included predictor and control variables. While we tentatively suggest that the associations observed here might be causal and the longitudinal design of the study facilitates this, the study is observational and hence we cannot rule out the presence of unmeasured confounders, which could be driving the associations that we present.

## Conclusion and implications

A high proportion of children do not meet expected levels of development in this ethnically diverse, socioeconomically disadvantaged population. Higher socioeconomic position may protect against poor early development in such communities, and the mechanism of this may differ by ethnic group. The high prevalence of children at risk can be partly explained by socioeconomic position, and this suggests a continued need for broader public health initiatives that meet the needs of families living in disadvantaged settings, such as proportionate universal approaches that encompass both universal prevention and early targeted interventions for those who need additional support.[6] This necessitates investment in early years practitioners, such as health visitors, who can work directly in these communities to identify children at risk and connect them with relevant support services.

**Social media** Kate E Mooney, LinkedIn @kate-mooney-27a1342a3

**Acknowledgements** The integration of research and practice in Bradford has only been possible because of the enthusiasm and commitment of staff and volunteers across children's services in Bradford. We are grateful to all Born in Bradford staff, the Better Start Bradford partnership and staff, all Better Start Bradford project teams, health professionals, local authority and voluntary and community sector organisations who have supported the integration of research into practice. We are grateful to all the families taking part in BiBBS and all members of the Community Research Advisory Group. A poster regarding this study was presented at the International Network for Research Inequalities in Child Health (INRICH) workshop in 2025.

**Contributors** KEM is responsible for the overall content as the guarantor. KEM conducted the analysis and drafted the manuscript. MW cleaned and linked the BiBBS baseline with the health visiting datasets. KEP, SLB, JL and JD all reviewed the manuscript and provided substantial contributions.

**Funding** This study has received funding from the National Lottery Community Fund (previously the Big Lottery Fund) as part of the A Better Start programme (Ref 10094849). The funder was not involved in the design of the study and collection, analysis and interpretation of data nor in writing the manuscript.SLB, JD and KP are also supported by the NIHR Yorkshire and Humber Applied Research Collaboration (ARC-YH; Ref: NIHR200166, see https://www.arc-yh.nihr.ac.uk). The views in this publication are those of the authors and not necessarily those of the NIHR or the Department of Health and Social Care.

**Competing interests** No, there are no competing interests.

**Patient and public involvement** Patients and/or the public were involved in the design, or conduct, or reporting, or dissemination plans of this research. Refer to the Methods section for further details.

**Patient consent for publication** Not applicable.

**Ethics approval** This study involves human participants and was approved by the protocol for BiBBS recruitment and collection of routine outcome data, which was approved by Bradford Leeds NHS Research Ethics Committee (15/YH/0455). Research governance approval was gained from Bradford Teaching Hospitals NHS Foundation Trust. The existing ethics includes approval for both evaluation of Better Start Bradford interventions and observational research. Hence, no further consent was sought for this study. Participants gave informed consent to participate in the study before taking part.

**Provenance and peer review** Part of a Topic Collection; Not commissioned; externally peer reviewed.

**Data availability statement** Data may be obtained from a third party and are not publicly available. Data cannot be shared publicly as they are available through a system of managed open access. Researchers interested in accessing the data can find details and procedures on the Born in Bradford website (https://borninbradford.nhs.uk/research/how-to-access-data/). Data access is subject to review by the Born in Bradford Executive, who review proposals on a monthly basis. Requests can be submitted to borninbradford@bthft.nhs.uk. Data sharing agreements are established between the researcher and the Bradford Institute for Health Research. Full analysis code and results are available at https://osf.io/c9rup.

**ORCID iDs**
Kate E Mooney https://orcid.org/0000-0003-4231-1643
Josie Dickerson https://orcid.org/0000-0003-0121-3406
Sarah Louise Blower https://orcid.org/0000-0002-9168-9995
Jennie Lister https://orcid.org/0000-0002-2911-8331
Kate E Pickett https://orcid.org/0000-0002-8066-8507

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
