## [Reviewer comments · BMJ Paediatrics Open]

ARTICLE DETAILS

TITLE (PROVISIONAL)	Do socioeconomic inequalities contribute to the high prevalence of child developmental risk in an ethnically diverse, socioeconomically disadvantaged population? A Born in Bradford's Better Start (BiBBS) study
AUTHORS	Mooney, Kate; Dickerson, Josie; Blower, Sarah Louise; Walker, Matthew; Lister, Jennie; Pickett, Kate

VERSION 1 - REVIEW

REVIEWER NAME	Neil Kaye
REVIEWER AFFILIATION	University College London
REVIEWER CONFLICT OF INTEREST	N/A
AI DISCLOSURE	No
DATE REVIEW RETURNED	18-Jul-2025

GENERAL COMMENTS	This is a well-researched, clearly-presented paper, which addresses a timely and under-researched aspect of socioeconomic inequalities within an ethnically diverse and deprived urban setting. I am pleased to be able to recommend it for publication. There are some very minor amendments that the authors may wish to make to improve the final version of the paper, outlined below: Abstract (p. 2) – There appears to be a word missing after “developmental” in line 27 Methods (p. 7) – You mention that ethnicity “is grouped into the largest ethnic groups in the cohort” and then proceed to list “White British or Irish”. I notice in the supplementary material that this group is labelled as White British. Please clarify if this group includes people of Irish ethnicity. Looking at the socio-cultural context around child development, it may be reasonable not to group Irish participants with the White British group. Discussion (p. 12) – when discussing “social gradients”, please clarify that you are using the gap between the highest and lowest categories and express these as ‘percentage points’ or ‘pp’, rather than using ‘%’. (p. 12) – Looking at the graphs, I am intrigued by the steepness of the gradient for the ‘Other’ ethnic group in relation to social support – especially compared to the other two categories. It would be good to include this in the discussion section as well.
--

REVIEWER NAME	Yu Wei Chua
REVIEWER AFFILIATION	University of Liverpool
REVIEWER CONFLICT OF INTEREST	N/A
AI DISCLOSURE	No
DATE REVIEW RETURNED	01-Aug-2025

GENERAL COMMENTS	Overall impression This manuscript presents important research questions and analyses that has the potential to inform understanding of the intersection between ethnic and socioeconomic inequalities. In particular, the authors present interesting questions and findings supported by robust statistical analysis on the role of social support in explaining ethnic differences in developmental outcomes from children from socioeconomically deprived backgrounds. This is an important area of research which can inform not just risk, but protective factors from the lens of ethnicity and culture. However, the justification and formulation of the research questions and the interpretation of the findings need significant revisions for this to come across. Major comments The authors frame the importance of the study in its consideration of subjective measures, specifically subjective measures of socioeconomic position. Beyond the common aspect that the measures are subjective/self-reported, there needs to be greater clarity on where social support sits in this. The distinction/similarities between the measures considered in the study are useful to outline, but what is key is the rationale for including each measure. The Millennium Cohort Study (MCS) is a similar study that has been designed to include measures of socioeconomic position (social class, family income, maternal education), alongside other finer grained / direct measures of specific family circumstances. This MCS briefing note (link below) distinguishes between measures of socioeconomic position and family circumstances well. https://cls.ucl.ac.uk/wp-content/uploads/2017/06/MCS-Age-11-initial-findings-Child-poverty-and-deprivation-briefing.pdf?_gl=1*onkixs*_up*MQ..*_ga*MTYwMTA1Njg4MC4xNzUzNzExODUz*_ga_EYRQV4V0KV*czE3NTM3MTE4NTEkbzEkZzAkdE3NTM3MTE4NTEkajYwJGwwJGgw In justifying the measures and informing the interpretations of the study, the authors may wish to consider frameworks and evidence for key pathways which socioeconomic position can influence child development, such as through financial circumstances (which can capture material deprivation and parental financial stress), parental behaviours, home learning environment. Cattan et al (ref below) provide a useful framework for understanding early childhood inequalities (material, emotional, educational). The authors should also consider which may be more be different in children from different ethnic backgrounds. Sarah Cattan, Emla Fitzsimons, Alissa Goodman, Angus Phimister, George B Ploubidis, Jasmin Wertz, Early childhood inequalities, Oxford Open Economics, Volume 3, Issue Supplement_1, 2024, Pages i711–i740, https://doi.org/10.1093/oec/odad072
---

Objective 1 appears like an afterthought compared to the other questions focusing on ethnic differences in socioeconomic inequalities. The authors could consider what are the limitations in current prevalence estimates that Objective 1 is addressing, that the Born in Bradford cohort is suitable for? For example, there are difficulties in routine collection of ethnic information in routine data, which makes it really hard to capture differences by ethnicity in routine data. Objective 1 could inform the prevalence rate in children from socioeconomically deprived groups, especially a breakdown by ethnicity, using the Born in Bradford cohort. If prevalence estimates are of interest, the authors should provide sampling information on the representativeness (potential biases due to drop out/missingness) as a cohort of children born in Bradford.

Introduction

- The first paragraph of the introduction captures the importance of early child development, but can focus more on the importance of the present study. This includes contextualising current understanding of ethnic inequalities in child development.
- The paragraph on socioeconomic position and child development can be summarised further – detail on all the specific measures not relevant but could be a point that the use of the Ages and Stages Questionnaire addresses methodological limitations in existing studies which focus on a limited domain of child development? Main point on the evidence across different cohort studies is that there is a socioeconomic gradient seen in various domains of child development, and using different measures?
- Rationale for looking at different measures of socioeconomic position needs refining, as this paragraph mentions mechanisms but the measures used in this study do not capture mechanisms. Could the point be that some measures of SEP may be more sensitive to specific mechanisms than others? E.g., maternal education may capture parental behaviours/ knowledge while income may capture material aspects. Then highlight the importance of subjective measures of socioeconomic status (e.g., more sensitive to basic needs).
- A more balanced presentation of the importance of, alongside the limitations, of subjective measures and objective measures is needed. Do your evidence Perhaps the point is that although there are limitations, subjective measures are important to complement objective measures?
- Social support isn't typically a measure of socioeconomic status, but a downstream factor important for health. Galobardes et al (2006) provide a useful definition of how an individual's SEP can be captured by various measures of the groupings and structures in society. Social support is also an important factor intersecting with ethnicity and culture. Perhaps the justification for the study is that subjective measures, including subjective social status and social support are important in investigating ethnic differences in socioeconomic deprivation?
- Language referring to ethnic differences: Needs better clarity whether the focus of the research on ethnic differences, or specifically ethnic minority groups compared to the majority group (literature review focuses on the latter). See ONS guidance on writing about ethnicity.
- There is ample evidence on ethnic differences in child development that can be presented here and the, e.g., from the Millennium Cohort Study – and if the focus is on ethnic differences, it is important to highlight the nuanced differences in child outcomes, as some ethnic

groups perform better compared to White children. In addition, Black Caribbean children tend to have worse outcomes compared to White British children, but Black African children perform better.

- Nice summary of the evidence and potential reasons why the effect of socioeconomic deprivation may be different in some ethnic groups. Are there specific ethnic groups of note here?
- Highlight the novelty of the BiBBS cohort and what gaps it was designed to address in research on child development/ inequalities?

Discussion

- P12 L9 “A high proportion” - Consider providing a comparison between the prevalence estimated in Born in Bradford (high % of ethnic minority, high deprivation cohort) to the prevalence of children not achieving GLD in other local sources, or in the general population, e.g., official statistics using the ASQ on fingertips?
- More balanced discussion of the strengths and limitations of assessing subjective social status needed.
- Conclusions: proportionate universalism came out of the blue. How does this relate to ethnicity?
- Could think about how interventions for socioeconomic inequalities can draw on social support?
- Important to discuss the key limitation in obtaining sufficiently sized ethnic groups in the Born in Bradford Cohort. The authors could consider what are the implications on gaps in understanding / interpretation of the present results? This is particularly relevant because the results show a specific role of social status for the South Asian group, but not the Other ethnic group. But it is conceivable that the Other ethnic group is hiding differences and similarities between South Asian and other groups in relation to the importance of social support. The authors could also discuss what are the ethnic groups they might expect to see similar findings (as the South Asian group), and therefore worth exploring more based on existing literature?
- Potential of missing data to bias the results also warrant discussion in the limitations and how they limit causal interpretations

Methods

- Missing data on ASQ need further description as there's substantial missing data – what are the demographics of children who did / did not have data on ASQ?

Minor comments

Abstract

- Results: Sufficient to indicate 22% out of 2,003 children at risk of poor development in the results, as “a high proportion” is repeated in the conclusion.
- Social status – clearer to refer to as subjective or self-reported social status, as maternal education measure of social status. Check that this is consistent throughout (e.g., in Introduction, subjective measures, it is also referred to as self-reported social status)

Methods

- Is this a cross-sectional/longitudinal study? In the discussion, the study is referred to as a longitudinal one.
- Methods were clearly described throughout and the authors are commended for considering unique confounder adjustment sets. Table 1 could probably be summarised into one or two sentences in text, and confounder adjustment sets indicated in Results tables in footnotes.

	 - P8, L34. Justification of use of the potentially at-risk cut-off? (currently a circular justification, using the potentially at risk cut-off “to create an outcome more sensitive to identifying children potentially at risk”. The authors could consider that there are ethnic differences in child development, and although the ASQ has been validated in several cultures/contexts, the same cut-off may not apply for children of all ethnicities. - P8, L34, FGLD method was specified to be used for Objective 3, but is also reported for Objective 2. - How was consent sought for data collection and linkage? - The authors may also state that no further consent was required for this secondary analysis Results  - Sample section needs some text - could have the Note in the main text and other key demographic descriptors of the Bradford cohort. - Some statistical estimates missing on P10 paragraph starting L50 which describes the figure? - Could state (shortened) objectives in headings to signpost the reader Highlights  - In general, could be more concise and needs greater focus on what is already known/what the study adds - Important to justify why we need to use different measures of socioeconomic status, and why ethnicity is important. This will also increase coherence with the implications on research, practice and policy section - Point 2 of What is already known - needs revision as it is a very long sentence presenting several important points on why the authors have considered several measures of socioeconomic status - Point 2 of What this study adds – clearer to refer to social status as subjective social status.
--	--

VERSION 1 – AUTHOR RESPONSE

Reviewer 1

Dr. Neil Kaye, University College London

Comment: This is a well-researched, clearly-presented paper, which addresses a timely and under-researched aspect of socioeconomic inequalities within an ethnically diverse and deprived urban setting. I am pleased to be able to recommend it for publication.

Response: Thank you Dr Neil Kaye for your considerate and helpful review, we are pleased that you recommend the paper for publication.

Comment: There are some very minor amendments that the authors may wish to make to improve the final version of the paper, outlined below:

Abstract

(p. 2) – There appears to be a word missing after “developmental” in line 27

Response: We have added the word ‘assessment’ (page 2).

Comment: (p. 7) – You mention that ethnicity “is grouped into the largest ethnic groups in the cohort” and then proceed to list “White British or Irish”. I notice in the supplementary material that this group is labelled as White British. Please clarify if this group includes people of Irish ethnicity. Looking at the socio-cultural context around child development, it may be reasonable not to group Irish participants with the White British group.

Response: Thank you for raising this. On reflection we agree with the Reviewer that White Irish participants should not be grouped together with the White British group. We have amended the methods in the manuscript, explaining that the White Irish participants are now grouped in Other group in the interaction models (changes made on p.7). We have also rerun the analysis with these new ethnic groups, hence we have amended the results tables with the new coefficients. The overall pattern of findings did not change (changes made on p.12 and Supplementary File D).

Comment: (p. 12) – when discussing “social gradients”, please clarify that you are using the gap between the highest and lowest categories and express these as ‘percentage points’ or ‘pp’, rather than using ‘%’.

Response: Thank you, we have amended the text to clarify that we are using percentage points (changes made on page. 13).

Comment: (p. 12) – Looking at the graphs, I am intrigued by the steepness of the gradient for the ‘Other’ ethnic group in relation to social support – especially compared to the other two categories. It would be good to include this in the discussion section as well.

Response: We have now expanded on this in the discussion by saying: “Interestingly, a very strong association was apparent between social support and child development for the ‘Other’ ethnic group; which includes White Other (including White Irish, Polish, Slovakian, Romanian, Czech, Other White, Gypsy/Roma/Irish Traveller), Black African/Caribbean/Black British, and Mixed ethnic groups. As these groups represent the least populous ethnic minorities in the BiBBS population, it may be that social support is more important for particular communities. Despite being an ethnic minority in England, the Pakistani population are the most populous ethnic group within the BiBBS area, perhaps meaning that they benefit from the psychosocial resilience offered by same-ethnic density, and that social support at a more individual level is not as crucial for improving child outcomes [44,45].” (change made on p.13-14).

Reviewer 2

Dr. Yu Wei Chua, University of Liverpool

Comments to the Author

Comment: Overall impression. This manuscript presents important research questions and analyses that has the potential to inform understanding of the intersection between ethnic and socioeconomic inequalities. In particular, the authors present interesting questions and findings supported by robust statistical analysis on the role of social support in explaining ethnic differences in developmental outcomes from children from socioeconomically deprived backgrounds. This is an important area of research which can inform not just risk, but protective factors from the lens of ethnicity and culture.

However, the justification and formulation of the research questions and the interpretation of the findings need significant revisions for this to come across.

Response: Thank you Dr. Yu Wei Chua for your acknowledgement of the importance of our study and thoughtful review of our manuscript. We appreciate your insights in our study and feel that the changes we have made outlined below have strengthened the manuscript.

The reviewed manuscript was total 3919 words, and the BMJ Paediatrics maximum word count is 4000. We therefore have requested from the Editors that we address the below comments using a more flexible word count, and we are pleased that they have agreed. The manuscript total word count is now 4904.

Comment: The authors frame the importance of the study in its consideration of subjective measures, specifically subjective measures of socioeconomic position. Beyond the common aspect that the measures are subjective/self-reported, there needs to be greater clarity on where social support sits in this. The distinction/similarities between the measures considered in the study are useful to outline, but what is key is the rationale for including each measure. The Millennium Cohort Study (MCS) is a similar study that has been designed to include measures of socioeconomic position (social class, family income, maternal education), alongside other finer grained / direct measures of specific family circumstances. This MCS briefing note (link below) distinguishes between measures of socioeconomic position and family circumstances well.

https://cls.ucl.ac.uk/wp-content/uploads/2017/06/MCS-Age-11-initial-findings-Child-poverty-and-deprivation-briefing.pdf?_gl=1*_onkixs*_up*MQ..*_ga*MTYwMTA1Njg4MC4xNzUzNzExODUz*_ga_EYRQV4V0KV*cZ3NTM3MTE4NTEkbzEkZzAkDE3NTM3MTE4NTEkajYwJGwwJGgw

Response: Thank you for raising this important point and allowing us the chance to clarify the consideration of social support. On reflection, we agree with the reviewer that social support is not strictly a measure of socioeconomic position, but can be considered alongside social capital as an aspect of people's social lives and social fabric, which is also related to socioeconomic position. We have therefore expanded upon this argument in our manuscript to now say: "Parental social support refers to the existence of social networks [21]. Although social support is not directly a measure of a family's socioeconomic position, it can be considered a specific manifestation of social capital, which refers to the resources that are obtained via a person's social networks [22, 23]. Parental social support could alleviate challenges during the transition to parenthood, hence supporting childhood outcomes [21], and could potentially mitigate against the negative effects of socioeconomic disadvantage for socioeconomically disadvantaged families." (change made on p. 5).

Comment: In justifying the measures and informing the interpretations of the study, the authors may wish to consider frameworks and evidence for key pathways which socioeconomic position can influence child development, such as through financial circumstances (which can capture material deprivation and parental financial stress), parental behaviours, home learning environment. Cattan et al (ref below) provide a useful framework for understanding early childhood inequalities (material, emotional, educational). The authors should also consider which may be more different in children from different ethnic backgrounds.

Sarah Cattan, Emla Fitzsimons, Alissa Goodman, Angus Phimister, George B Ploubidis, Jasmin Wertz, Early childhood inequalities, Oxford Open Economics, Volume 3, Issue Supplement_1, 2024, Pages i711–i740, <https://doi.org/10.1093/ooec/odad072>

Response: Thank you for raising this. We have expanded our paragraph explaining how socioeconomic position may impact children's development, and included reference to this important study. We have included a sentence about social status here, as this helps to address one of the Reviewer's later comments.

Specifically, we now say that: "Socioeconomic position may impact children's development through multiple mechanisms, for instance, socioeconomic advantage may afford parents better resources for enhancing the home environment of children, including via educational inputs such as play and stimulation, and emotional inputs such as parent-child relationships and interaction [12]. Socioeconomic advantage may also provide higher education for enhancing child development through cognitive stimulation and more complex language use. On the other hand, socioeconomic disadvantage may have a negative impact upon children's development through poor parental mental health and stress impacting upon parental engagement [12–14]. Subjective social status captures a parent's perception of their social class relative to other people, and may impact on children's outcomes through a heightened experience of chronic stress responses [15]. It is therefore of interest to compare the magnitude of associations for multiple measures, including not only traditional measures such as income and education, but also subjective aspects of societal position [16]. This can give insights into the mechanisms by which family socioeconomic position influences children's development [17]." (changes made on p.4)

Comment: Objective 1 appears like an afterthought compared to the other questions focusing on ethnic differences in socioeconomic inequalities. The authors could consider what are the limitations in current prevalence estimates that Objective 1 is addressing, that the Born in Bradford cohort is suitable for? For example, there are difficulties in routine collection of ethnic information in routine data, which makes it really hard to capture differences by ethnicity in routine data. Objective 1 could inform the prevalence rate in children from socioeconomically deprived groups, especially a breakdown by ethnicity, using the Born in Bradford cohort. If prevalence estimates are of interest, the authors should provide sampling information on the representativeness (potential biases due to drop out/missingness) as a cohort of children born in Bradford.

Response: We appreciate the reviewers point that the Born in Bradford's Better Start (BiBBS) cohort could be used to estimate the prevalence of poor childhood development by ethnic group, and this is an area of research we'd like to pursue in the future. However, this is not the purpose of the current study. This would involve a much wider literature review and discussion around the contextual factors and possible explanations for potential differences in child development by ethnic group, which we feel exceeds the scope of the current study.

We have now mentioned the information regarding sample representativeness: "An interim BiBBS cohort profile for mothers recruited between 2016 and 2020 was shown to be broadly representative of the pregnancy population in terms of ethnicity, parent age, parity, language ability, and area deprivation [32]." (change made on p.7).

Comment: Introduction

- The first paragraph of the introduction captures the importance of early child development, but can focus more on the importance of the present study. This includes contextualising current understanding of ethnic inequalities in child development.

Response: As stated above, we feel that a consideration of differences in child development by ethnic group is beyond the scope of the current study. Our manuscript includes a summary of two studies

which show the inequalities in development that ethnic minority children in the UK face on page 5, which we feel is sufficient to introduce the literature that contextualises our third research objective (socioeconomic gradients within ethnic groups).

Comment: - The paragraph on socioeconomic position and child development can be summarised further – detail on all the specific measures not relevant but could be a point that the use of the Ages and Stages Questionnaire addresses methodological limitations in existing studies which focus on a limited domain of child development?

Response: Thank you for raising the important point that using the ASQ addresses a previous methodological limitation in research. We have now stated “Building on previous research which has tended to focus on one specific aspect of child development at a time, we use the Ages and Stages Questionnaire, which is a broad measure of child development that encompasses communication, fine and gross motor skills, problem solving, and personal-social development” (change made on p. 6).

Comment: Main point on the evidence across different cohort studies is that there is a socioeconomic gradient seen in various domains of child development, and using different measures?

We agree that this is the intended main point, we have therefore amended a sentence to say “The majority of the relevant literature demonstrating pervasive socioeconomic inequalities in early child development in the UK is based on cohort studies, across various socioeconomic exposures and multiple domains of child development [7]” (change made on p. 4).

Comment: - Rationale for looking at different measures of socioeconomic position needs refining, as this paragraph mentions mechanisms but the measures used in this study do not capture mechanisms. Could the point be that some measures of SEP may be more sensitive to specific mechanisms than others? E.g., maternal education may capture parental behaviours/ knowledge while income may capture material aspects. Then highlight the importance of subjective measures of socioeconomic status (e.g., more sensitive to basic needs).

Response: Thank you for helping us to elaborate this point, this is what we were trying to suggest. We have amended the text throughout the introduction to make this clearer, specifically around subjective social status giving an insight into stress (changes made on p.4).

Comment: - A more balanced presentation of the importance of, alongside the limitations, of subjective measures and objective measures is needed. Do your evidence Perhaps the point is that although there are limitations, subjective measures are important to complement objective measures?

Response: Yes, the point is that subjective measures of socioeconomic position may be important in income deprived populations. We believe this has been addressed by the previous comment, and by clarifying our inclusion of social support as a measure (see response to comment above).

Comment: - Social support isn't typically a measure of socioeconomic status, but a downstream factor important for health. Galobardes et al (2006) provide a useful definition of how an individual's SEP can be captured by various measures of the groupings and structures in society. Social support is also an important factor intersecting with ethnicity and culture. Perhaps the justification for the study is that subjective measures, including subjective social status and social support are important in investigating ethnic differences in socioeconomic deprivation?

Response: Thank you, we have responded to your comment about social support above. On your point regarding the justification for the study, we agree, and we have clarified this argument about using a variety of socioeconomic measures in other ethnic groups: “Subjective measures of socioeconomic position may aid our understanding of relationships between socioeconomic position and child outcomes for ethnic minority groups, as they may be more relevant than traditional

measures (e.g. education), and, again, may provide an insight into the mechanisms by which minority families mitigate against the negative effects of disadvantage” (change made on p.5).

Comment: - Language referring to ethnic differences: Needs better clarity whether the focus of the research on ethnic differences, or specifically ethnic minority groups compared to the majority group (literature review focuses on the latter). See ONS guidance on writing about ethnicity.

Response: In response to an earlier comment, we have clarified that the current manuscript is not about ethnic differences in children’s outcomes, as this would exceed the scope of the current study. We have searched the manuscript for the phrase ‘ethnic differences’ and ensured that we have not mentioned ethnic differences.

We have examined the ONS guidance on writing about ethnicity and feel that we are consistent with this. Although we use slightly different ethnic groupings to the recommended groups, our groupings in our study better represent the Bradford population (e.g. we specifically use South Asian to group Pakistani, Bangladeshi, and Indian ethnicities, as these are the most populous groups in our cohort). If there is a specific part of the ONS guidance you feel we have not followed, please let us know.

Comment - There is ample evidence on ethnic differences in child development that can be presented here and the, e.g., from the Millennium Cohort Study – and if the focus is on ethnic differences, it is important to highlight the nuanced differences in child outcomes, as some ethnic groups perform better compared to White children. In addition, Black Caribbean children tend to have worse outcomes compared to White British children, but Black African children perform better.

Response: Whilst we agree this is an important research area of interest, we feel that a focus on ethnic differences in child development exceeds the scope of our study, please see above for our response on this.

Comment: - Nice summary of the evidence and potential reasons why the effect of socioeconomic deprivation may be different in some ethnic groups. Are there specific ethnic groups of note here?

Response: We have added the specific ethnic groups (change made on p.5).

Comment: - Highlight the novelty of the BiBBS cohort and what gaps it was designed to address in research on child development/ inequalities?

Response: We have amended a paragraph to say “The present study uses data from the Born in Bradford’s Better Start (BiBBS) interventional birth cohort, which has recruited a representative sample of 5,758 pregnant mothers and their children living in three inner-city areas of Bradford. BiBBS recruited families throughout the time period that ‘Better Start’ interventions were being delivered (1st January 2016 to 31st July 2024). Whilst BiBBS was specifically designed to undertake efficient and pragmatic evaluations of these multiple early life interventions, the cohort data can also be used for understanding how inequalities develop in disadvantaged populations.” (change made on p.6)

Comment: Discussion: - P12 L9 “A high proportion” - Consider providing a comparison between the prevalence estimated in Born in Bradford (high % of ethnic minority, high deprivation cohort) to the prevalence of children not achieving GLD in other local sources, or in the general population, e.g., official statistics using the ASQ on fingertips?

Response: The prevalence estimate for England is already provided after this sentence, in line 10 (see line 10, p. 12). As outlined in our methods, this estimate was obtained from a national study using ASQ data (page. 8).

Comment: - More balanced discussion of the strengths and limitations of assessing subjective social status needed.

Response: Thank you for giving us the opportunity to expand this argument. We have now stated that “Since this measure of subjective social status purportedly captures a person’s perception of their social class relative to other people, this may indicate that a parent’s perceptions of inequality has an impact on their children’s outcomes through a heightened experience of chronic stress [15,44]. Since few epidemiological studies include both subjective and objective measures of socioeconomic position, further research is necessary. We propose that subjective socioeconomic measures may be particularly useful in ethnically diverse populations through potentially obtaining lower non-response rates compared to objective metrics (like exact income or foreign qualifications) that can be challenging to reliably report. Further research is needed regarding the validity and generalisability of socioeconomic measures across different ethnic groups, and their relationships with various outcomes” (change made on p.13).

Comment: - Conclusions: proportionate universalism came out of the blue. How does this relate to ethnicity?

Response: The recommendation for a proportionate universal approach stems from the finding that a very high proportion of children are at risk of poor child development in our study, and that both universal prevention and targeted support will be required to meet the needs of all families. We have amended the text to make this clearer: “The high prevalence of children at risk can be partly explained by socioeconomic position, and this suggests a continued need for broader public health initiatives that meet the needs of families living in disadvantaged settings, such as proportionate universal approaches that encompass both universal prevention and early targeted interventions for those who need additional support [6].” (change made on p. 15)

Comment: - Could think about how interventions for socioeconomic inequalities can draw on social support?

Response: Thank you for raising this. This is an important point and we appreciate the opportunity to expand on this. We have now added the following paragraph: “Altogether, the strong relationship between subjective social status and child outcomes may have implications for early interventions that support families facing socioeconomic disadvantage. This may suggest that early parenting interventions should not only provide educational material and financial aid to families, but also address the psychological impacts associated with low social status [44]. This indicates the importance of the social environment for parents of young children, highlighting the role that group-based parenting programmes could play in building social networks and social cohesion for parents of young children [48]. However, as we found that the importance of social status and support may vary by ethnic group, such interventions may work well for parents of some ethnicities, but not for others. Indeed, evidence for the effectiveness of early parenting interventions is limited in minority groups [49].” (change made on p.14).

Comment: - Important to discuss the key limitation in obtaining sufficiently sized ethnic groups in the Born in Bradford Cohort. The authors could consider what are the implications on gaps in understanding / interpretation of the present results? This is particularly relevant because the results show a specific role of social status for the South Asian group, but not the Other ethnic group. But it is conceivable that the Other ethnic group is hiding differences and similarities between South Asian and other groups in relation to the importance of social support. The authors could also discuss what are the ethnic groups they might expect to see similar findings (as the South Asian group), and therefore worth exploring more based on existing literature?

Response: There is a very large sample of ethnic minorities recruited in the BiBBS cohort, as the majority of the sample in the interim cohort profile were found to be from ethnic minority groups (88%)

(Dickerson et al., 2020). We have now added this as a strength to our manuscript to ensure this is clear (change made on p.14).

In contrast to what the Reviewer has said here, the results show a significant role for social status for both White British and Other ethnic groups, with a weaker relationship for South Asian groups (see “The adjusted predictions (see below) indicate that social status and support were less strongly related to child development for South Asian groups, in comparison to White British and Other ethnic groups”, page 12).

We are unsure how to address the Reviewer’s last point regarding similar findings to the South Asian group – but we would welcome any further suggestions on this from them.

Comment: - Potential of missing data to bias the results also warrant discussion in the limitations and how they limit causal interpretations

Comment: Methods: - Missing data on ASQ need further description as there’s substantial missing data – what are the demographics of children who did / did not have data on ASQ?

Response: Thank you for highlighting this. We have added a section regarding missing data to the strengths and limitation section: “Some families were missing a baseline questionnaire (around 25%) due to changes to recruitment procedures during COVID, and some children were missing the ASQ outcome (28%, see Figure 1). Hence our complete case analysis is only valid under the assumption having complete data does not depend on a child’s development, after taking into account the included predictor and control variables.” (change made on p.14).

Comment: Abstract: - Results: Sufficient to indicate 22% out of 2,003 children at risk of poor development in the results, as “a high proportion” is repeated in the conclusion.

Response: Thank you, we have made this change.

Comment: - Social status – clearer to refer to as subjective or self-reported social status, as maternal education measure of social status. Check that this is consistent throughout (e.g., in Introduction, subjective measures, it is also referred to as self-reported social status)

Response: Thank you, we have made this change throughout, apart from in the abstract due to word count limitations.

Comment: Methods: - Is this a cross-sectional/longitudinal study? In the discussion, the study is referred to as a longitudinal one.

Response: This is a longitudinal study. We have now amended the study design text to explicitly say it is longitudinal, and mentioned that the outcome is collected approximately 2-2.5-years post birth (changes made on page 7).

Comment: - Methods were clearly described throughout and the authors are commended for considering unique confounder adjustment sets. Table 1 could probably be summarised into one or two sentences in text, and confounder adjustment sets indicated in Results tables in footnotes.

Response: Thank you for acknowledging the unique confounder adjustment sets. We considered this suggestion, but we have left Table 1 as a table as we feel that provides explicit clarity on which variables are confounders/covariates and feel that this written into text would quickly become repetitive. We have now included the confounder adjustment sets as notes attached to our results tables in Supplementary Material C and D.

Comment: - P8, L34. Justification of use of the potentially at-risk cut-off? (currently a circular justification, using the potentially at risk cut-off “ to create an outcome more sensitive to identifying children potentially at risk”. The authors could consider that there are ethnic differences in child development, and although the ASQ has been validated in several cultures/contexts, the same cut-off may not apply for children of all ethnicities.

Response: Thank you for giving us the opportunity to explain this better. We also agree that the ASQ cut-offs may not apply for children of all ethnicities and have included a sentence on its psychometric properties: “Several studies have shown the ASQ-3 to be a valid screening tool to identify developmental concerns for individual children, though there is a lack of its validation in England specifically, and for ethnic minority groups” (changes made on p.8).

Regarding the justification of the Full GLD method, we have now amended this to say “The FGLD method provides more variation in the outcome than the GLD method, which means it is more sensitive to identifying any existing socioeconomic gradients in outcomes.” Indeed, we find that the ‘Full’ GLD method allows detection of areas where children may be at risk of poor development, identifying a further 439 children with potential developmental concerns. (changes made on p.8).

Comment: - P8, L34, FGLD method was specified to be used for Objective 3, but is also reported for Objective 2.

Response: We have now amended the text to say “For objective 1 we described the number of children at risk of poor development overall using both the GLD and FGLD method. For reference we compare proportions of children at risk using the GLD method to Jung et al. (2025), who described child developmental outcomes in the ASQ-3 across England.” (changes made on p.8).

Comment: - How was consent sought for data collection and linkage?

Response: The recruitment and consent process is outlined under the ‘study setting’ heading (see p. 7). We have made minor clarifications to this text to clarify that recruitment took place in person, and that participants provided written consent for both data collection and linkage (changes made on p.7).

Comment: - The authors may also state that no further consent was required for this secondary analysis

Response: We have now stated in our ethical approval section that “The existing ethics includes approval for both evaluation of Better Start Bradford interventions, and observational research. Hence, no further consent was sought for this study” (changes made on p.15).

Comment: Results: - Sample section needs some text - could have the Note in the main text and other key demographic descriptors of the Bradford cohort.

Response: We have now added text to this section explaining the figure, and moved the note into the main text (changes made on p.11).

Comment: - Some statistical estimates missing on P10 paragraph starting L50 which describes the figure?

Response: We have stated that full models are provided in a Supplementary file (due to a maximum of tables being allowed in BMJ publications), and the key statistical estimates are given in the paragraph presented prior to the figure, where it says “Odds ratios from adjusted logistic regressions between the predictor variables and achieving the expected level of development show that having a degree (OR=2.09, 95% CI 1.33 to 3.30), living comfortably (OR=1.90,95% CI 1.08 to 3.36), and social status (OR=1.13, 95%CI 1.03 to 1.24) all resulted in increased odds of having a FGLD. Social support was not associated with having a FGLD (OR=1.01, 95% CI 0.95 to 1.07).”

Comment: - Could state (shortened) objectives in headings to signpost the reader

Response: Thank you, we have done this (changes made to p.11-12).

Comment: Highlights: - In general, could be more concise and needs greater focus on what is already known/what the study adds

Response: Thank you, in addressing the below comments we feel the highlights section has improved. We have also made the sentences more concise, and have added a sentence on our new section regarding the implications for interventions.

Comment: - Important to justify why we need to use different measures of socioeconomic status, and why ethnicity is important. This will also increase coherence with the implications on research, practice and policy section

Response: Thank you, we have now mentioned 'within ethnically diverse samples', and clarified the importance of the subjective measures.

Comment: - Point 2 of What is already known - needs revision as it is a very long sentence presenting several important points on why the authors have considered several measures of socioeconomic status

Response: Thank you, we have broken this into two sentences.

Comment: - Point 2 of What this study adds – clearer to refer to social status as subjective social status.

Response: Thank you, we have done this.

VERSION 2 - REVIEW

REVIEWER NAME	Neil Kaye
REVIEWER AFFILIATION	University College London
REVIEWER CONFLICT OF INTEREST	N/A
AI DISCLOSURE	No
DATE REVIEW RETURNED	20-Oct-2025

GENERAL COMMENTS	Thank you for addressing all my concerns in relation to the paper. I am now delighted to recommend that it be published.
--

REVIEWER NAME	Yu Wei Chua
REVIEWER AFFILIATION	University of Liverpool
REVIEWER CONFLICT OF INTEREST	N/A
AI DISCLOSURE	No
DATE REVIEW RETURNED	07-Nov-2025

GENERAL COMMENTS	Many thanks to the authors for considering my comments and responding to these in detail. I feel my comments have mostly been adequately addressed and I just have some clarifications which may require further adjustments (on the approach to examine ethnic groups, and on ethnic differences in the association of social support and child development), and a few minor comments on the presentation of added text.
--

Ethnic groupings

The terminology used to refer to different ethnic groups and ethnic minority groups is appropriate. To clarify, my view is that the authors could include a nuanced argument around the differences between ethnic groups. Although the authors do not use the term BAME, the reason that BAME is no longer an appropriate term is that it obscures differences between different ethnic groups (<https://equalities.blog.gov.uk/2022/04/07/why-we-no-longer-use-the-term-bame-in-government/#:~:text=%E2%80%98EE%80%80BAME%EE%80%81%E2%80%99%20is%20frequently>). Although the term ethnic minority group is used appropriately (ONS <https://www.ethnicity-facts-figures.service.gov.uk/style-guide/writing-about-ethnicity/>), the authors could elaborate on differences between specific ethnic groups, especially in relation to the research area of interest and the BiB cohort context. There is also general guidance on ethnic groupings according to 2021 Census (<https://www.ethnicity-facts-figures.service.gov.uk/style-guide/ethnic-groups/>), although in this study the grouping of South Asian ethnic groups together is justifiable, despite Pakistani being the largest ethnic group in the BiB catchment area.

More specifically, first, the authors could include some details in the introduction to build the rationale behind examining differences in SEP between different ethnic groups – this would improve the language in describing about ethnic minority groups to show recognition of the ethnic diversity amongst ethnic minorities in the UK. For example, in Line 50, you mentioned discrimination and racism, but there are also specific experiences of families of Pakistani ethnicity that could be more relevant to the exposures considered here (social support).

Second, the authors should also include a thoughtful justification of why they have chosen to group the different ethnic groups, in addition to the methodological constraint of small group sizes. This is particularly important if Aim 3 of the study is truly to “Investigate if the association... varies by ethnic group”, and not just a focus on comparing ethnic minority groups as a whole to the majority White ethnic group. In particular, a line on the justification of groupings of Pakistani, Indian and Bangladeshi ethnic groups together could be included, especially because Pakistani appears to be the majority ethnic group in the BiB cohort and the Other South Asian ethnic group is not much smaller than the White British group. For example, it could be justifiable given the focus on social support and social capital, of which Pakistani, Indian and Bangladeshi ethnic groups have similar cultural practices in relation to social support, multigenerational households. Although it is a balance of methodology and aims, clarifying this focus is also important in justifying why the ethnic groups are collapsed in this way in the study (e.g., why African, Chinese, Caribbean and White Irish are grouped together in the Other category).

Third the limitation in grouping all other ethnic minority groups together (White Other, Chinese, African Caribbean, Mixed, or Other) should also be discussed. This could provide a more balanced view on which aspect of your findings are truly generalisable as suggested in P14 L47-48 (“Findings from BiBBS may generalise to many similar urban populations, especially those with high ethnic

diversity and socioeconomic deprivation. The large number of ethnic minorities recruited (e.g. n=979 Pakistani, 49% of the sample) improves our statistical power to detect differences and builds on previous research, which may have been limited by smaller numbers of ethnic minorities”).

Additionally, in response to Reviewer 1’s comment the authors interpret the stronger association between social support and child development in the Other group to the fact that they are least populous (P14 L3 As these groups represent the least populous ethnic minorities in the BiBBS population, it may be that social support is more important for particular communities, and P14 L12: Our study extends this by demonstrating that a subjective measure of social support may have different relationships with children’s outcomes depending on ethnic group, and may be particularly important for less populous ethnic minorities). The importance of ethnic density influencing the benefits of social support statement should perhaps be rephrased as a point for further research in relation to the Other group. A number of reasons: a. although the association is stronger in the Other group, the confidence intervals is much wider for those reporting High social support (Figure 4D), this suggests there is potentially more variation in the effect of high social support on child development in the Other group (as it is a highly heterogeneous group). b. there could also be fewer individuals reporting High social support in the Other group. but the authors haven’t provided the distribution of the socioeconomic exposures between the different ethnic groups; c. although the research cited does implicate ethnic density (Pickett et al) and evidence shows that higher ethnic density is protective for Pakistani and Indian mothers, but ethnic density was in fact unrelated to infant and maternal outcomes in Black African and Black Caribbean mothers. Therefore the authors have perceptively captured the potential benefit of ethnic density for the Pakistani population, how ethnic density (being less populous) affects the Other ethnic groups could be rephrased for clarity. For example the authors could recognise there are several potential mechanisms that ethnic density may explain their findings in the Other group (e.g., assimilation, integration, alongside prejudice/stigma), but caution that this requires further research which ethnic groups falling under the Other ethnic group may be driving the stronger relationship of social support on child development due to the diverse sociocultural experiences in the Other group.

Social support

Referring to the authors’ response: In contrast to what the Reviewer has said here, the results show a significant role for social status for both White British and Other ethnic groups, with a weaker relationship for South Asian groups (see “The adjusted predictions (see below) indicate that social status and support were less strongly related to child development for South Asian groups, in comparison to White British and Other ethnic groups”, page 12).

This is interesting and I wonder what the variation in Social support and Subjective Social Status is for the South Asian ethnic group compared to the White British, and Other Ethnic group. The supplementary material only provides the descriptive statistics of each exposure characteristic by outcome status, but it would also be useful to provide the descriptive statistics of all the exposures by ethnicity to see the distribution of the exposures. Is social support

already generally high with smaller variation in the South Asian group?

In addition, in the Discussion p14, L16 the authors discuss limitations in the measure of social support in terms of limitations capturing the quality and type of support over just the quantity of social connections, and critically highlight that South Asian families are characterised by close proximity to family members/ living in the same household. The authors however, could be more explicit in how “the nature and dynamics of this support might differ” (p14 L24). Could this be in relation to child rearing practices within multigenerational families might buffer the negative impact of low social support on child development? Given the existing research suggest a less pronounced or lack of socioeconomic gradients in maternal and child health outcomes, is there also research supporting the idea that multigenerational child rearing practices may buffer against negative outcomes due to certain parental-level socioeconomic attributes? The authors also highlight evidence that ethnic density is protective - and could include information on how ethnic density offers psychosocial protection through shared culture, social networks, social capital (Pickett et al., 2009)

As the explanation behind why the association of social support on child development is weaker in South Asian groups is at present speculative, it might be premature to suggest that social status and support is not important for some ethnic groups or that parenting (P15 L34. “However, as we found that the importance of social status and support may vary by ethnic group, such interventions may work well for parents of some ethnicities, but not for others. Indeed, evidence for the effectiveness of early parenting interventions is limited in minority groups [49]”)

Minor comments on presentation

Introduction

Pg 4 Para starting L35 – there is some repetition in the examples provided in the first sentence and subsequent statements.

L38 – “Socioeconomic advantage may also provide higher education for enhancing child development...”

This sentence can be rephrased potentially with a reference to differences in the home learning or educational environment (which the first sentence has already captured).

L42 – Could be more explicit what “parental engagement” means, perhaps in relation to parent-child relationships?

L44 - “traditional” measures (and in other sections of the paper) could imply that the measurement / definition of SEP needs to be updated. – perhaps more accurate to refer to income and education as “objective” measures?

Pg 4 L42-44 – “Subjective social status captures a parent’s perception of their social class relative to other people, and may impact on children’s outcomes through a heightened experience of chronic stress responses [15]”

There are also other limitations of objective measures, which subjective measures can address - which you touch upon later in the paragraph on ethnic inequalities.

	Pg 4 Para starting L15 - Having read the added paragraph following from this para, it seems appropriate to identify that the gap in the existing work on SEP and child development is that these have largely focused on objective measures of SEP – education, employment, occupation. But examining / comparing associations using different measures of objective and subjective SEP and associated concepts (e.g., social capital) may inform subtle differences in how SEP influences children’s development. Section on Ethnicity and Child development – the authors have clarified that the scope of study focuses on the association between SEP and child development more broadly with ethnic differences in SEP one of the aims (which show results worth exploring further in future). The reason I felt that examining SEP and ethnic inequalities were one of the key aims is because the selection of SEP indicators appear to be driven by this specific aim on ethnic and cultural difference. Perhaps as a transition sentence moving from the section on subjective measures to ethnic group, the authors could consider stating that that subjective measures of socioeconomic position can be complementary to objective measures is in examining the socioeconomic gradient in child development in different ethnic groups. Discussion Paragraph on early interventions: P14 L 28-30. This may suggest that early parenting interventions should not only provide educational material and financial aid to families, but also address the psychological impacts associated with low subjective social status [44] - The authors restrict the discussion to parenting interventions / programmes, but financial aid does not necessarily fall under parenting. Perhaps it is more accurate to rephrase to mention broadly the different approaches needed. E.g., different pathways in which socioeconomic disadvantage impact child development likely require different approaches to effectively address, including access to educational materials, financial aid, support for parental stress, and social support such as through group-based parenting programmes. The authors can also mention the role of integrated family hubs in addressing complex needs faced by some families from low socioeconomic backgrounds.
--	---

VERSION 2 – AUTHOR RESPONSE

Reviewer: 1

Dr. Neil Kaye, University College London

Comments to the Author

Thank you for addressing all my concerns in relation to the paper. I am now delighted to recommend that it be published.

Reviewer: 2

Dr. Yu Wei Chua, University of Liverpool

Comments to the Author

Many thanks to the authors for considering my comments and responding to these in detail. I feel my comments have mostly been adequately addressed and I just have some clarifications which may require further adjustments (on the approach to examine ethnic groups, and on ethnic differences in the association of social support and child development), and a few minor comments on the presentation of added text.

Ethnic groupings

The terminology used to refer to different ethnic groups and ethnic minority groups is appropriate. To clarify, my view is that the authors could include a nuanced argument around the differences between ethnic groups. Although the authors do not use the term BAME, the reason that BAME is no longer an appropriate term is that it obscures differences between different ethnic groups ([https://www.ethnicity-facts-figures.service.gov.uk/style-guide/writing-about-ethnicity/](https://equalities.blog.gov.uk/2022/04/07/why-we-no-longer-use-the-term-bame-in-government/#:~:text=%E2%80%98EE%80%80BAME%EE%80%81%E2%80%99%20is%20frequently). Although the term ethnic minority group is used appropriately (ONS ), although in this study the grouping of South Asian ethnic groups together is justifiable, despite Pakistani being the largest ethnic group in the BiB catchment area.

More specifically, first, the authors could include some details in the introduction to build the rationale behind examining differences in SEP between different ethnic groups – this would improve the language in describing about ethnic minority groups to show recognition of the ethnic diversity amongst ethnic minorities in the UK. For example, in Line 50, you mentioned discrimination and racism, but there are also specific experiences of families of Pakistani ethnicity that could be more relevant to the exposures considered here (social support).

Response: Thank you for the clarification on this, we have now expanded this argument in our introduction to say:

“Relevant to the sample included in this study, South Asian families parenting styles may differ to other ethnic groups, and South Asian families are more likely to involve intergenerational or extended support in parenting itself [32–34]. The relationship between socioeconomic experiences and children’s outcomes may therefore differ between ethnic groups, and this relationship may be mitigated or exacerbated by different cultural practices, such as parenting.” [page 5]

Second, the authors should also include a thoughtful justification of why they have chosen to group the different ethnic groups, in addition to the methodological constraint of small group sizes. This is particularly important if Aim 3 of the study is truly to “Investigate if the association... varies by ethnic group”, and not just a focus on comparing ethnic minority groups as a whole to the majority White ethnic group. In particular, a line on the justification of groupings of Pakistani, Indian and Bangladeshi ethnic groups together could be included, especially because Pakistani appears to be the majority ethnic group in the BiB cohort and the Other South Asian ethnic group is not much smaller than the

White British group. For example, it could be justifiable given the focus on social support and social capital, of which Pakistani, Indian and Bangladeshi ethnic groups have similar cultural practices in relation to social support, multigenerational households. Although it is a balance of methodology and aims, clarifying this focus is also important in justifying why the ethnic groups are collapsed in this way in the study (e.g., why African, Chinese, Caribbean and White Irish are grouped together in the Other category).

Response: Thank you for highlighting this and encouraging us to justify this analytic decision. We have now said in our methods:

“South Asian ethnicities were grouped as one due to potential cultural similarities in parenting style amongst these groups [31–33], and similarities in historic migration patterns [39]. We grouped the ‘Other’ ethnicities and retain them as (1) the minority groups were too small to estimate separately, and (2) such groups have historically been excluded from research and analysis [40]. Hence, although the ‘Other’ group contain multiple heterogenous ethnic groups with a variety of experiences, including their data may allow for an insight into patterns in their outcomes, and generate further hypothesis testing with larger samples.” [page 8].

Third the limitation in grouping all other ethnic minority groups together (White Other, Chinese, African Caribbean, Mixed, or Other) should also be discussed. This could provide a more balanced view on which aspect of your findings are truly generalisable as suggested in P14 L47-48 (“Findings from BiBBS may generalise to many similar urban populations, especially those with high ethnic diversity and socioeconomic deprivation. The large number of ethnic minorities recruited (e.g. n=979 Pakistani, 49% of the sample) improves our statistical power to detect differences and builds on previous research, which may have been limited by smaller numbers of ethnic minorities”).

Response: Thank you for this. We believe this is partly addressed by addressing the comment above, and we have added this to the limitations in the discussion section to say:

“The large number of Pakistani families recruited (e.g. n=979, 49% of the sample) improves our statistical power to detect differences and builds on previous research describing outcomes for this ethnic minority group, which may have previously been limited by smaller numbers of ethnic minorities. Whilst it is a strength that our sample on the whole is ethnically diverse, we were limited in our ability to describe associations between socioeconomic position and child outcomes within each of these groups due to small group sizes.” [page 15].

Additionally, in response to Reviewer 1’s comment the authors interpret the stronger association between social support and child development in the Other group to the fact that they are least populous (P14 L3 As these groups represent the least populous ethnic minorities in the BiBBS population, it may be that social support is more important for particular communities, and P14 L12: Our study extends this by demonstrating that a subjective measure of social support may have different relationships with children’s outcomes depending on ethnic group, and may be particularly important for less populous ethnic minorities).

The importance of ethnic density influencing the benefits of social support statement should perhaps be rephrased as a point for further research in relation to the Other group. A number of reasons: a. although the association is stronger in the Other group, the confidence intervals is much wider for those reporting High social support (Figure 4D), this suggests there is potentially more variation in the effect of high social support on child development in the Other group (as it is a highly heterogeneous group). b. there could also be fewer individuals reporting High social support in the Other group. but the authors haven't provided the distribution of the socioeconomic exposures between the different ethnic groups; c. although the research cited does implicate ethnic density (Pickett et al) and evidence shows that higher ethnic density is protective for Pakistani and Indian mothers, but ethnic density was in fact unrelated to infant and maternal outcomes in Black African and Black Caribbean mothers. Therefore the authors have perceptively captured the potential benefit of ethnic density for the Pakistani population, how ethnic density (being less populous) affects the Other ethnic groups could be rephrased for clarity. For example the authors could recognise there are several potential mechanisms that ethnic density may explain their findings in the Other group (e.g., assimilation, integration, alongside prejudice/stigma), but caution that this requires further research which ethnic groups falling under the Other ethnic group may be driving the stronger relationship of social support on child development due to the diverse sociocultural experiences in the Other group.

Response: Thank you for this, we agree it is important to reflect on the limitation regarding the diverse Other ethnic group and the implications for our study findings. We have now said: "However, it is crucial to acknowledge that the 'Other' ethnic group in this study are heterogenous and hence there may be hidden variation across ethnicities in the association between social support and child outcomes. Hence further research is needed with these smaller ethnic minorities to explore these associations in depth." [page 14].

Social support

Referring to the authors' response: In contrast to what the Reviewer has said here, the results show a significant role for social status for both White British and Other ethnic groups, with a weaker relationship for South Asian groups (see "The adjusted predictions (see below) indicate that social status and support were less strongly related to child development for South Asian groups, in comparison to White British and Other ethnic groups", page 12).

This is interesting and I wonder what the variation in Social support and Subjective Social Status is for the South Asian ethnic group compared to the White British, and Other Ethnic group. The supplementary material only provides the descriptive statistics of each exposure characteristic by outcome status, but it would also be useful to provide the descriptive statistics of all the exposures by ethnicity to see the distribution of the exposures. Is social support already generally high with smaller variation in the South Asian group?

Response: Thank you, we have added an additional table to our supplementary descriptive statistics file which describes each exposure and covariate by ethnicity. South Asian groups on average rate their social support as higher than the Other ethnic group, but similarly to the White British group, and that the variation in social support is fairly consistent across groups.

In addition, in the Discussion p14, L16 the authors discuss limitations in the measure of social support in terms of limitations capturing the quality and type of support over just the quantity of social connections, and critically highlight that South Asian families are characterised by close proximity to family members/ living in the same household. The authors however, could be more explicit in how “the nature and dynamics of this support might differ” (p14 L24). Could this be in relation to child rearing practices within multigenerational families might buffer the negative impact of low social support on child development? Given the existing research suggest a less pronounced or lack of socioeconomic gradients in maternal and child health outcomes, is there also research supporting the idea that multigenerational child rearing practices may buffer against negative outcomes due to certain parental-level socioeconomic attributes?

The authors also highlight evidence that ethnic density is protective - and could include information on how ethnic density offers psychosocial protection through shared culture, social networks, social capital (Pickett et al., 2009)

Response: Thank you, we think this is an interesting point. We have not been able to identify other relevant research that suggests multigenerational child rearing may buffer against negative child outcomes, hence have added a tentative statement to say “These strong familial networks could potentially buffer against the negative effects of socioeconomic disadvantage, though other research in this area is lacking.” [page 14].

In relation to ethnic density, we have added that South Asian groups may “benefit from the psychosocial resilience offered by same-ethnic density through shared culture, networks, and social capital.” [page 14].

As the explanation behind why the association of social support on child development is weaker in South Asian groups is at present speculative, it might be premature to suggest that social status and support is not important for some ethnic groups or that parenting (P15 L34. “However, as we found that the importance of social status and support may vary by ethnic group, such interventions may work well for parents of some ethnicities, but not for others. Indeed, evidence for the effectiveness of early parenting interventions is limited in minority groups [49]”)

Response: We have amended this to say “However, as we found that the importance of subjective social status and support may vary by ethnic group, further research on this topic is needed with diverse ethnic groups.” [page 14].

Minor comments on presentation

Introduction

Pg 4 Para starting L35 – there is some repetition in the examples provided in the first sentence and subsequent statements.

Response: We have now removed a sentence here to avoid repetition and integrated it into the overall sentence (see below).

L38 – “Socioeconomic advantage may also provide higher education for enhancing child development...”

This sentence can be rephrased potentially with a reference to differences in the home learning or educational environment (which the first sentence has already captured).

Response: In response to the Reviewer’s previous comment, we have removed this sentence to avoid repetition and integrated it with the first one. We now say:

“Socioeconomic position may impact children’s development through multiple mechanisms, for instance, socioeconomic advantage may afford parents better resources and education for enhancing the home environment of children, including via educational inputs such as language, play and stimulation, and emotional inputs such as parent-child relationships and interaction” [page 4].

L42 – Could be more explicit what “parental engagement” means, perhaps in relation to parent-child relationships?

Response: We have amended this to say: “stress impacting upon a parent’s engagement and relationship with their child”. [page 4]

L44 - “traditional” measures (and in other sections of the paper) could imply that the measurement / definition of SEP needs to be updated. – perhaps more accurate to refer to income and education as “objective” measures?

Response: We have amended “traditional” to “objective” throughout.

Pg 4 L42-44 – “Subjective social status captures a parent’s perception of their social class relative to other people, and may impact on children’s outcomes through a heightened experience of chronic stress responses [15]”

There are also other limitations of objective measures, which subjective measures can address - which you touch upon later in the paragraph on ethnic inequalities.

Response: We agree with the Reviewer that there are limitations of objective measures and that we do describe these in our later section on ethnicity. We feel that this description is placed appropriately as it is.

Pg 4 Para starting L15 - Having read the added paragraph following from this para, it seems appropriate to identify that the gap in the existing work on SEP and child development is that these have largely focused on objective measures of SEP – education, employment, occupation. But examining / comparing associations using different measures of objective and subjective SEP and associated concepts (e.g., social capital) may inform subtle differences in how SEP influences children's development.

Response: We have added a sentence to explicitly identify this gap, saying "Despite this, the majority of the previously summarised literature applies objective measures of socioeconomic position such as education, occupation, or income [8–11]." [page 4].

We agree with the Reviewer that comparing different objective and subjective socioeconomic measures may inform differences in how SEP influences child development, and this is at present stated where we say the following on Page 4:

"Subjective social status captures a person's perception of their social class relative to other people, and may impact on children's outcomes through a heightened experience of chronic stress responses [15]. It is therefore of interest to compare the magnitude of associations for multiple measures, including not only traditional objective measures such as income and education, but also subjective aspects of societal position [16]. This can give insights into the mechanisms by which family socioeconomic position influences children's development [17]."

Section on Ethnicity and Child development – the authors have clarified that the scope of study focuses on the association between SEP and child development more broadly with ethnic differences in SEP one of the aims (which show results worth exploring further in future). The reason I felt that examining SEP and ethnic inequalities were one of the key aims is because the selection of SEP indicators appear to be driven by this specific aim on ethnic and cultural difference. Perhaps as a transition sentence moving from the section on subjective measures to ethnic group, the authors could consider stating that that subjective measures of socioeconomic position can be complementary to objective measures is in examining the socioeconomic gradient in child development in different ethnic groups.

Response: Thank you for clarifying. We appreciate the Reviewer's suggestion but feel that contextualising ethnic inequalities in child development is an important first step prior to explaining socioeconomic gradients within ethnic groups. We attempted to amend the text in the way suggested but found it difficult to build towards the argument that we make in the following paragraph, hence we have retained the current structure in this section. We hope the Reviewer can accept this as a minor difference in writing styles.

Discussion

Paragraph on early interventions: P14 L 28-30. This may suggest that early parenting interventions should not only provide educational material and financial aid to families, but also address the psychological impacts associated with low subjective social status [44]

- The authors restrict the discussion to parenting interventions / programmes, but financial aid does not necessarily fall under parenting. Perhaps it is more accurate to rephrase to mention broadly the different approaches needed. E.g., different pathways in which socioeconomic disadvantage impact child development likely require different approaches to effectively address, including access to educational materials, financial aid, support for parental stress, and social support such as through group-based parenting programmes. The authors can also mention the role of integrated family hubs in addressing complex needs faced by some families from low socioeconomic backgrounds.

Response: This paragraph makes the argument that parenting interventions should not only provide educational material and financial aid, but also address psychological impacts of low social status. We do feel this section broadly summarises the argument the Reviewer makes here, and we apologise that we are unsure of what exact changes are being suggested.

However, we'd like to address that whilst the Reviewer states that 'financial aid does not necessarily fall under parenting', we do know that parenting interventions do, on occasion, encompass financial aid/assistance, and this is based on our in-depth knowledge of the interventions that the families in this cohort may have been exposed to. Perhaps this helps to clarify our argument.

VERSION 3 - REVIEW

REVIEWER NAME	Yu Wei Chua
REVIEWER AFFILIATION	University of Liverpool
REVIEWER CONFLICT OF INTEREST	N/A
AI DISCLOSURE	No
DATE REVIEW RETURNED	02-Dec-2025

GENERAL COMMENTS	Many thanks to the authors for addressing each comment and explaining their position in detail. I'm pleased to recommend this manuscript for publication.
---